# Improving Thermal Conductivity and Tribological Performance of Polyimide by Filling Cu, CNT, and Graphene

**DOI:** 10.3390/mi14030616

**Published:** 2023-03-07

**Authors:** Chen Liu, Jingfu Song, Gai Zhao, Yuhang Yin, Qingjun Ding

**Affiliations:** 1State Key Laboratory of Mechanics and Control of Mechanical Structures, Nanjing University of Aeronautics and Astronautics, Nanjing 210016, China; 2School of Naval Architecture & Ocean Engineering, Jiangsu Maritime Institute, Nanjing 211170, China; 3State Key Laboratory of Bio-Fibers and Eco-Textiles, Qingdao University, Qingdao 266071, China

**Keywords:** polyimide, copper, carbon nanotubes, graphene nanosheets, molecular dynamics simulation

## Abstract

The thermal conductivity, mechanical, and tribological properties of polyimide (PI) composites filled by copper (Cu), carbon nanotube (CNT), graphene nanosheet (GNS), or combination were investigated by molecular dynamics simulation (MD). The simulated results suggested that Cu can improve thermal stability and thermal conductivity, but it reduces mechanical properties and tribological properties. CNT and GNS significantly improved the thermal and tribological properties at low content, but they decreased the properties at high content. In this study, the modification mechanism, friction, and wear mechanism of different fillers on polyimide were revealed by observing the frictional interface evolution process from the atomic scale, extracting the atomic relative concentration, the temperature and velocity distribution at the friction interface, and other microscopic information.

## 1. Introduction

Polyimide (PI) composites with good mechanical properties, heat resistance, and tribological properties are ideal frictional materials for ultrasonic motors [1]. However, the thermal conductivity of polyimide is only 0.30 W/m·K. It is easy to generate heat accumulation at the friction interface under a high vacuum environment, which seriously limits its application in the space environment.

Modification is one of the most effective ways to improve the thermal conductivity of PI. Compared with polymer materials, metals such as copper (Cu), silver (Ag), gold (Au), and other heavy metals, as well as carbon-based materials, such as carbon nanotubes (CNT) and graphene (GN), have higher thermal conductivity. Yang et al. [2] used carbon particles and SiO_2_ to improve the thermal conductivity of PI. They also found that both silicon and carbon could improve the thermal conductivity. Wu et al. [3] used boron nitride nanosheets (BNNSs) and silver nanoparticle (AgNPs) hybrid-filled polyimide to improve the thermal conductivity of PI. The results showed that, with the increase in the content of hybrid filler, the thermal conductivity of the composite also increased, and it was significantly higher than that of BNNSs/PI composite. Gong et al. [4] prepared graphene fabric with mesh structure by chemical vapor precipitation method to improve the thermal conductivity of PI. The results showed that, after adding 10 layers of graphene woven fabrics (about 12 wt.%), the thermal conductivity of the composite could be increased by 1418% compared with pure PI. He et al. [5] studied the thermal conductivity of copper to polyoxymethylene (POM) composites by using a hot disk analyzer, and they studied the tribological properties by using a M-2000 friction and wear test machine. The results showed that 3 wt.% copper particles had little effect on the thermal conductivity of POM composites, but with the increase in copper content, the thermal conductivity of the composites would increase. Friction coefficient and wear rate will increase accordingly. Carbon nanotubes are also often used to improve the thermal conductivity of materials due to their excellent thermal conductivity [6,7,8]. Smith et al. [9] also studied the effects of nano-filled materials, mainly steel nanospheres, carbon nanotubes, and graphene sheets, on the thermal conductivity of fluoropolymers. The results showed that carbon nanotubes and graphene could effectively improve the thermal conductivity of polymers. In addition, many scholars have improved GNS to achieve better application [10,11,12]. Yang et al. [13] studied the friction effect of carbon nanotubes with different functions on composites through molecular dynamics simulation. The simulation results show that proper functionalization of carbon nanotubes can enhance the interface bonding between carbon nanotubes and matrix materials, thus improving the mechanical properties of the composite and reducing the wear rate. Cai et al. [14] also found that adding a small amount of graphene nanosheets could synergically enhance the thermal conductivity of low dielectric constant boron nitride polytetrafluoroethylene composites. Although the principle of adding high thermal conductivity filler to low thermal conductivity PI to improve the thermal conductivity [15] is widely recognized, no one has studied the effects of copper, carbon nanotubes, and graphene on the thermal conductivity changes of polyimide, and they have not studied the tribological properties at the same time.

This paper selected copper as a filler based on the ultrasonic motor copper stator, and copper also has high thermal conductivity and low cost. Except for copper, carbon nanotubes and graphene were added to improve the thermal conductivity, mechanical properties, and wear resistance of PI. Due to a large number of additive types, the filling ratio of each type was optimized by molecular dynamics simulation to reduce the time and cost of experimental research. With the development of the computer level, molecular dynamics simulation has become a very mature and reliable method and has been widely used in scientific research. It can observe the intrinsic mechanism of mixtures at the atomic level and predict the properties of mixtures.

## 2. Materials and Methods

### 2.1. Establishment of Molecular Dynamics Model of Composite Materials

The original structure of PI is designed as shown in Figure 1a. The copper clusters are then imported from the structure library of molecular dynamics simulation software Material Studio (Version 2019), as is shown in Figure 1b. Carbon nanotubes with the size of 25.3 Å × 8.3 Å and two-dimensional graphene nanosheet models with the size of 21.15 Å × 21.18 Å are established, as shown in Figure 1c,d.

Periodic boundary condition cubic units with the size of 4.0 × 4.0 × 4.0 nm^3^ are constructed, and then the PI molecular chains with the degree of polymerization of 2 were filled by the Monte Carlo rule [16]. After molecular dynamics optimization, the packing density is close to the actual density, which is 1.6 g/cm^3^. Finally, PI molecules were filled into the periodic cell, as shown in Figure 2a. The three modifiers shown in Figure 1 are filled into the pure PI model, the composite models of Cu/PI, CNT/PI, and GNS/PI formed are shown in Figure 2, and the graphene and carbon nanotubes were colored green for easy identification. In order to compare the effects of modifiers more clearly, a set of composite models with three modifiers added to PI were also designed, as shown in Figure 2e.

### 2.2. Model Optimization

The Condensed-Phase Optimized for Atomistic Simulation Studies (COMPASS) [17] force field, which is suitable for simulating polymer systems, was selected in all optimization processes. Ewald and Atom methods were used to analyze van der Waals and Coulomb interactions between PI, Cu, CNT, and GNS [18,19,20]. The detailed parameters in the optimization process are shown in Table 1. Firstly, the Smart method was used for geometric optimization, and then the global minimum energy configuration was obtained. To further relax the molecular chains of the model, a 15-week annealing process was performed. In order to obtain a sufficient stable and reasonable model, the lowest energy model was selected from 15 annealing cycles for dynamic optimization. The isothermal ensemble system was used for dynamic optimization at atmospheric pressure 10^−4^ GPa and time step 1 fs.

### 2.3. Calculation of Thermal and Tribological Properties

Thermal conductivity is a measure of a substance’s ability to transfer heat through a material by conduction. It is measured by the length of the material under unit temperature difference and unit time direct conduction of heat, and usually λ is used to represent the thermal conductivity. The thermal conductivity is calculated according to the following, Figure 3 [21]:

In order to better simulate the actual operation of the ultrasonic motor, the friction model, as shown in Figure 4, is established, and the friction model was established after matching with copper pairs. The pressure on the surface of PI nanocomposites is tested by pressing a 10 Å × 10 Å × 7.2 Å copper nano pin. The bottom layer is a 40 Å × 40 Å × 7.2 Å copper atomic layer. Before friction, the temperature was set to 298 K, and NVT ensemble was selected for all the friction simulations. The tribological properties of the friction materials were analyzed by running 300 ps at 0.1 Å/ps under the loading of 0.01 GPa to obtain the trajectory file with time change.

## 3. Results and Discussion

### 3.1. Thermal Properties

The PI composites with three different components are designed for each composite. In Cu/PI, the content of Cu powder is 3 wt.%, 6 wt.%, and 9 wt.%, respectively. The contents of carbon nanotubes and graphene in CNT/PI and GNS/PI are set at 0.5 wt.%, 1 wt.%, and 1.5 wt.%, which took into account the fact that carbon nanomaterials are extremely easy to aggregate, leading to the degradation of mechanical properties [22,23,24].

After adding Cu to PI, the thermal conductivity of the composite was improved, and the thermal conductivity increased with the copper content, as shown in Figure 5a. However, the thermal conductivity increases little when the Cu content is small. The thermal conductivity of 3 wt.% Cu/PI increases by 5.3% compared with pure PI. When the copper content reaches 9 wt.%, the thermal conductivity has a qualitative leap, reaching 0.521 W/m·K. The increase is 34.5% relative to pure PI. This is because, when Cu content is high, it is easy to form a heat conduction path, so the thermal conductivity is greatly improved [25,26,27].

Meanwhile, the mechanical properties of Cu/PI composites are calculated, and the results are shown in Figure 5b. The mechanical properties of the composites decreased after Cu was added. When the Cu content is 9 wt.%, Young’s modulus and shear modulus of PI are 2.62 GPa and 1.21 GPa, which decreased by 20.6% and 24.4%, respectively. This is because, with the increase in copper content, the interaction force between polymers of PI composite materials is weakened, which makes copper powder particles easily cause stress concentration phenomenon and promotes the formation of cracks.

After the addition of CNT, the thermal conductivity of PI composite increased first and then decreased, as shown in Figure 6a. The thermal conductivity of PI composite reached the peak of 0.88 W/m·K at the CNT content of 1.0%, which was 131.6% higher than pure PI. This is because of the composite effect [28]. When CNTs are at the stage of low mass fraction, the thermal conductivity of the composite increases with the increase in the content of the filler of CNT. At this point, the thermal conductivity of CNT plays a leading role in the overall thermal conductivity of the composites. With the excessive increase in CNT concentration, the stable arrangement of CNT in the composite material is damaged, the distribution of CNT become disordered, the normal heat conduction mode is broken, and the direction and path of heat conduction begin to be disordered, reducing the thermal conductivity of aerogel. 

Young’s modulus and shear modulus are also tested, as shown in Figure 6b. When the CNT with 0.5 wt.% mass fraction is added, Young’s modulus and shear modulus of the PI composite reach the maximum, which is 3.71 MPa, increased by 12.4%, and the shear modulus reaches 1.92 MPa, increased by 20%. This is because, when the stress is applied along the axial direction of CNT, the matrix can transfer the stress to the axial direction of CNT through intermolecular force, which enhances the structure of the composite material and improves the hardness of the composite material. In the simulation, the content of CNT is adjusted by changing the length of CNT. Moreover, the length of CNT increased, resulting in the disconnection of PI [29]. As a result, the mechanical properties decrease with the increase in the content of CNT.

Figure 7a shows the thermal conductivity of GNS/PI composites. When the GNS content was 1 wt.%, the in-plane thermal conductivity reached 0.44 W/m·K, which increased by 15.8%. The enhancement effect of GNS on the thermal conductivity of PI is determined by the specific surface area. However, the graphene nanosheet is a two-dimensional planar structure. Both sides can contact PI molecules and can be embedded into the matrix in a flat way, which undoubtedly increases the probability of receiving phonon transport [30].

Both the shear modulus and Young’s modulus of the GNS/PI composite in Figure 7b decrease with the increase in GNS content. This is because GNS itself has very high mechanical properties. However, with the increase in GNS content, the mechanical properties began to decline. This is mainly because the larger area of the graphene makes the polymer discontinuous.

Regarding the selection of materials, we should not only consider the thermal properties, but also the mechanical properties of materials. Therefore, a set of optimal amounts of modifiers are set to be mixed together, namely, PI composites containing 3 wt.% Cu, 0.5 wt.% CNT, and 0.5 wt.% GNS. The comparison of thermal conductivity of all materials is shown in Figure 8. Compared with pure PI and Cu/PI, the carbon material had better thermal conductivity. The 0.5 wt.% CNT/PI had the highest thermal conductivity, with 1.02 W/m·K. Unfortunately, the combination one did not show high thermal performance. With regard to the analysis from their structure in Figure 4e, the heat conduction channel did not form between Cu, CNT, and GNS due to steric hindrance. Therefore, the thermal conductivity of the composite is not as high as that of CNT/PI and GNS/PI. In future studies, the fillers’ proportions and structure distribution will focus and be designed to further improve their thermal and mechanical performance.

### 3.2. Tribological Properties

The friction coefficient (COF) and wear rate under the optimal addition amount of modifier are obtained through the molecular dynamics simulation calculation, as shown in Figure 9. The friction coefficient after adding copper is 16.7% higher than that of pure PI because copper has a higher hardness in the polymer matrix, and it is not easy to change its form or position under the action of shear or extrusion [31]. Compared with pure PI, the friction coefficient of the added CNT decreased by 23.3%. The reduction was the same with the addition of GNS modifier. This is due to the strong adsorption between the CNT and the PI matrix, which can cluster PI molecules and reduce the adhesion on the copper friction pair, thus reducing the friction coefficient. This is similar to the principle of GNS. When three modifiers were added to the PI, the friction coefficient decreased even more.

In order to further observe the dynamical evolution of Cu, CNT, and GNS in the wear state of PI matrix, the state diagram of the model after the end of friction is extracted, as shown in Figure 10. By comparing the deformation degree of the models in Figure 10, it can be found that the deformation degree of pure PI is the largest, while the composite model with Cu, CNT, and GNS is relatively small due to its better mechanical properties. Additionally, the deformation degree is minimal when Cu, CNT, and GNS are added to the PI simultaneously.

Then, the changes of temperature and energy in the friction process are also analyzed to explore the mechanism of friction reduction. First, the atomic relative concentration distribution along the z direction of the friction PI nanocomposite is extracted, as shown in Figure 11. The relative atomic concentration of the four models at the bottom contact surface is high, indicating that there is a strong interaction between the polymer molecules and the copper layer. Compared with pure PI, the peak relative concentration after the addition of modifiers is smaller. The peak relative concentration of Cu, CNT, and GNS is reduced by 58.6%, 41.3%, and 27.5%, respectively. The peak relative concentration of three modifiers is reduced by 51.7%. This is because, after adding the modifier, there will be adsorption between the modifier and PI, so the relative atomic concentration of the interface is lower than that of pure PI. After 300 ps of friction, some of the molecules adhered to the copper pin, so the concentration decreased.

Then, the distribution diagram of atomic temperature along the direction of matrix thickness in the process of molecular dynamics friction simulation is calculated, as shown in Figure 12. This occurs at 54 Å along the thickness of the matrix, which is the friction interface between PI matrix and copper pin layer, where a peak temperature of 333 K appeared. In contrast, the peak temperature of the matrix with modifier added at the friction interface is about 300 K. The frictional interface temperatures of Cu, CNT, and GNS are reduced by 9.6%, 10.2%, and 9.9%, respectively. When three modifiers are added at the same time, the friction interface temperature decreased to 295 K, which was 11.4% lower than the pure PI. It should be pointed out that this phenomenon is consistent with the conclusion of atomic concentration distribution in Figure 10. Higher interface friction temperature not only easily damages the viscous properties of the polymer, but it also easily produces adhesive wear, leading to reduced service life.

As shown in Figure 13, the pure PI matrix also has an obvious peak velocity of 0.9 Å/ps at the friction interface, while the matrix with the modifier does not have an obvious peak. In the direction of thickness, the movement speed of Cu and CNT is relatively slow, and there is only a small peak at 11 Å, which is the friction interface between the friction pair and the composite material. The motion velocity fluctuation of the composite material with GNS added is relatively large, so the motion velocity fluctuation of the composite material with three modifiers is also relatively obvious, but they do not exceed the peak value of pure PI. This phenomenon is also consistent with the results of atomic concentration distribution and friction interface temperature. Fewer molecular chains interact with the copper pin, reducing temperature of the friction interface. Therefore, the movement of atoms is affected by the temperature, which limits the atomic movement and reduces the potential wear amount, thus enhancing the tribological properties of the polymer matrix.

## 4. Conclusions

The effects of copper powder, carbon nanotube, and graphene on the thermal conductivity of PI are studied by molecular dynamics simulation. The conclusion is as follows:Copper can improve the thermal conductivity of PI, but at the same time, due to the uneven distribution of copper, it also reduces the mechanical properties of PI. Additionally, the hardness of copper is very large, which directly increases the friction coefficient of composite materials.Carbon nanotube (CNT) and graphene can improve the performance of PI very well. The 0.5 wt.% mass fraction of carbon nanotubes can, respectively, increase the axial thermal conductivity by 115.8%, and the maximum thermal conductivity of graphene in the two-dimensional plane direction can increase by 168.4%. Both of them can effectively reduce the friction and wear of the composites and make the composites have excellent tribological properties. However, the strength of the composite decreases as the content of carbon nanotubes and graphene continues to increase.The composites with three fillers at the same time have no obvious change in the increase in thermal conductivity, but they can greatly reduce the friction coefficient of the composites and reduce the wear rate.

## Figures and Tables

**Figure 1 micromachines-14-00616-f001:**
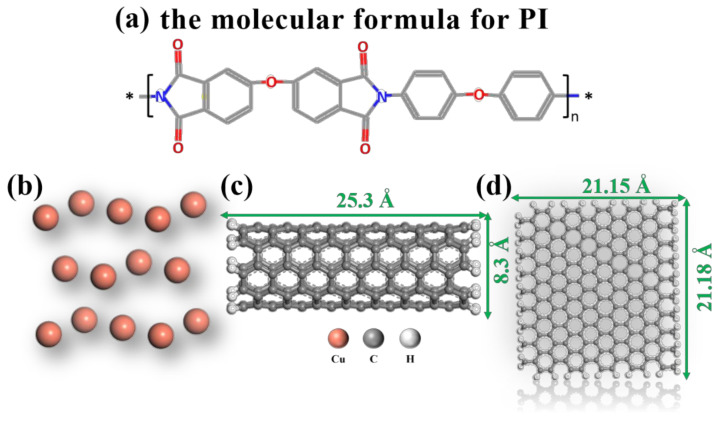
Structural information of (**a**) PI, (**b**) Cu, (**c**) CNT, and (**d**) GNS.

**Figure 2 micromachines-14-00616-f002:**
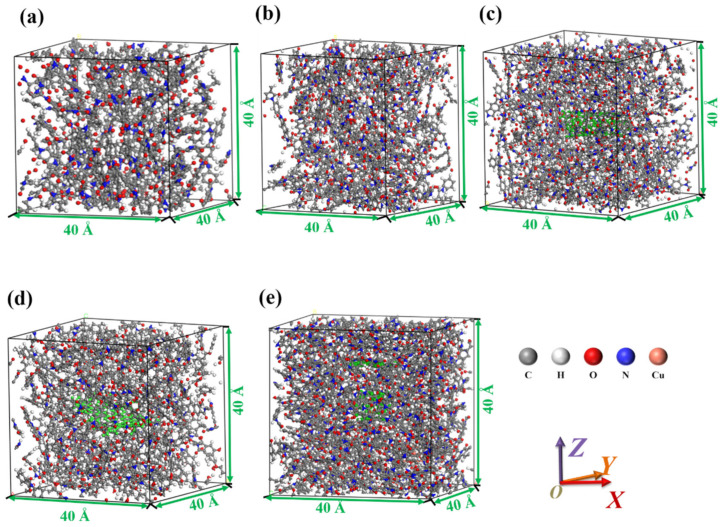
Molecular models of the (**a**) PI, (**b**) Cu/PI, (**c**) CNT/PI, (**d**) GNS/PI, and (**e**) composite structure.

**Figure 3 micromachines-14-00616-f003:**
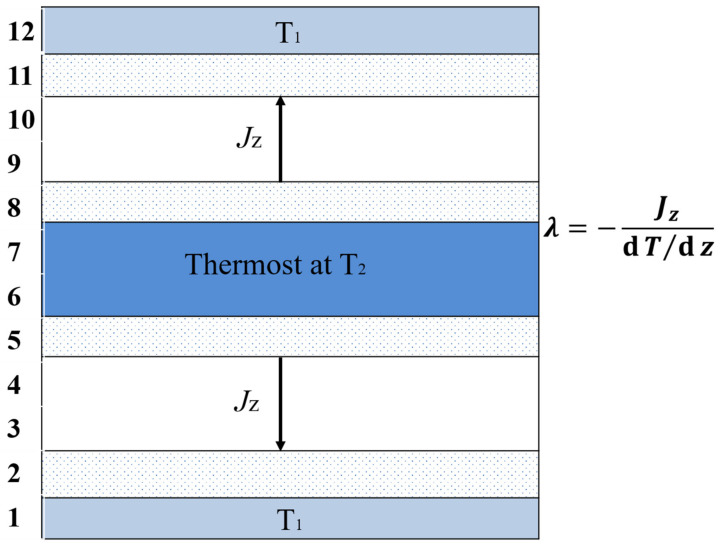
Schematic diagram of calculation of thermal conductivity.

**Figure 4 micromachines-14-00616-f004:**
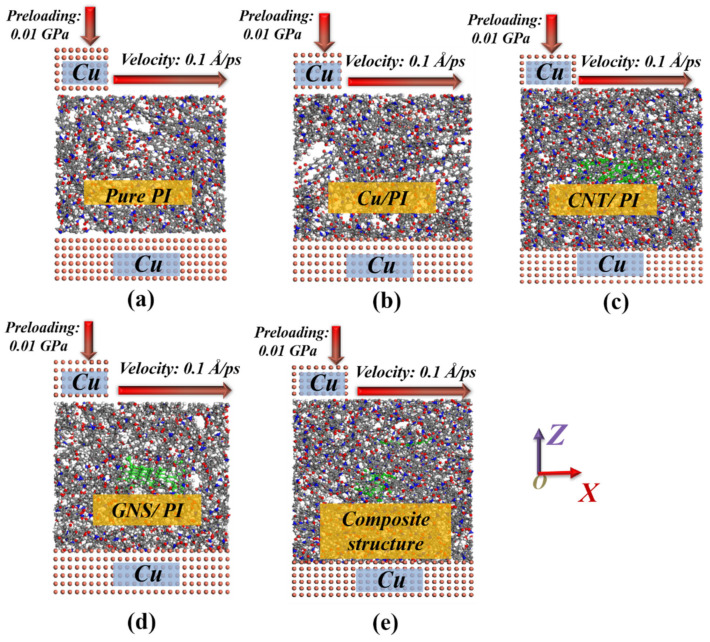
Friction pair model of the (**a**) PI, (**b**) Cu/PI, (**c**) CNT/PI, (**d**) GNS/PI, and (**e**) composite structure.

**Figure 5 micromachines-14-00616-f005:**
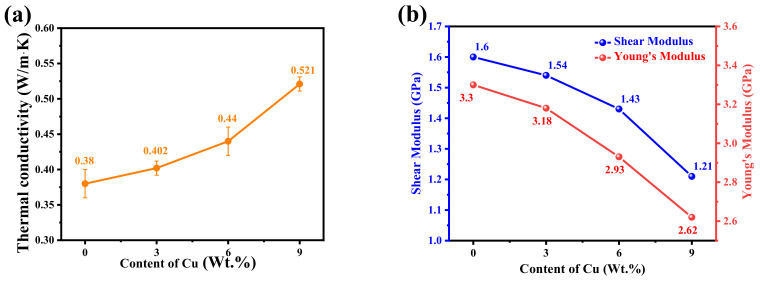
(**a**) Thermal conductivity, (**b**) Young’s Modulus and Shear Modulus of Cu/PI.

**Figure 6 micromachines-14-00616-f006:**
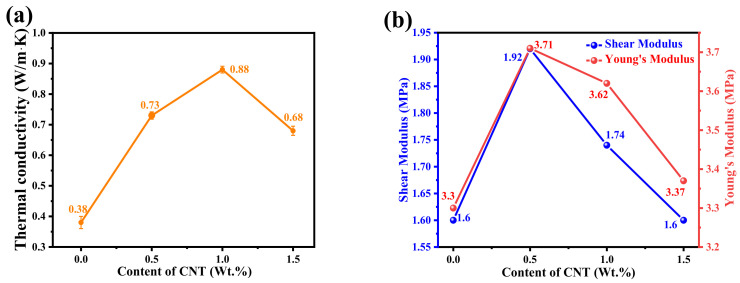
(**a**) Thermal conductivity, (**b**) Young’s Modulus and Shear Modulus of CNT/PI.

**Figure 7 micromachines-14-00616-f007:**
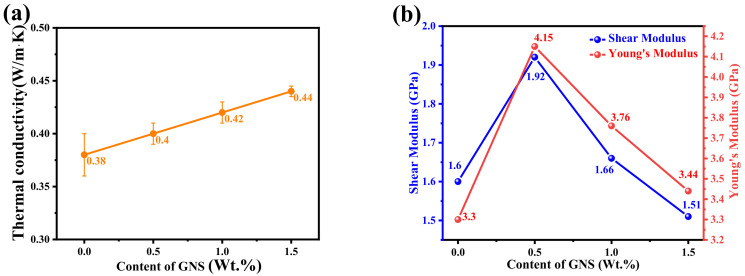
(**a**) Thermal conductivity, (**b**) Young’s Modulus and Shear Modulus of GNS/PI.

**Figure 8 micromachines-14-00616-f008:**
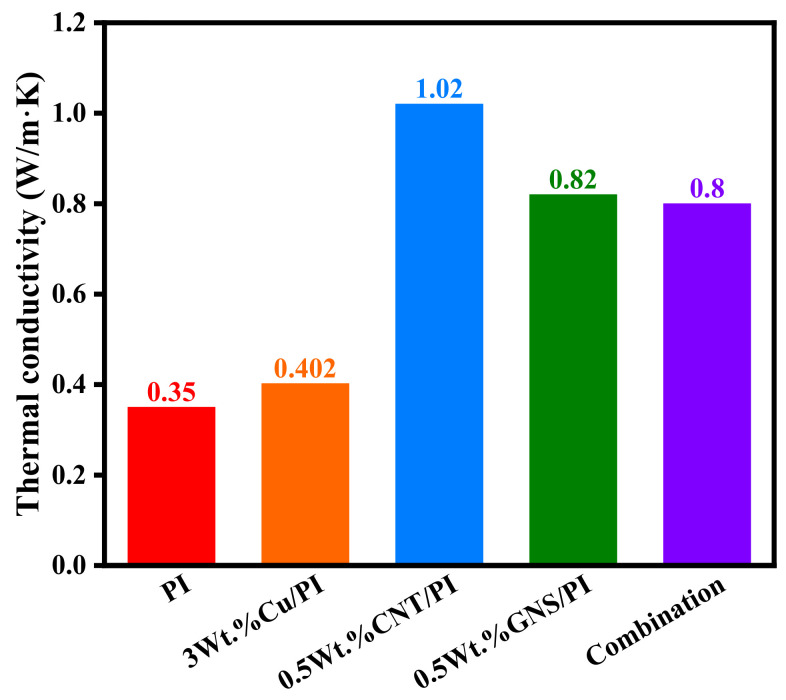
Thermal conductivity of molecular models.

**Figure 9 micromachines-14-00616-f009:**
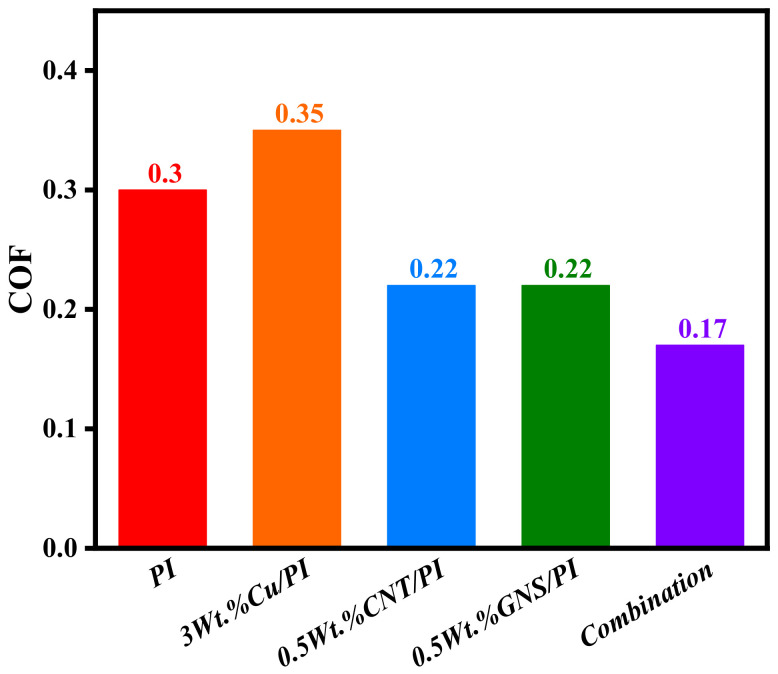
Friction coefficient of friction models.

**Figure 10 micromachines-14-00616-f010:**
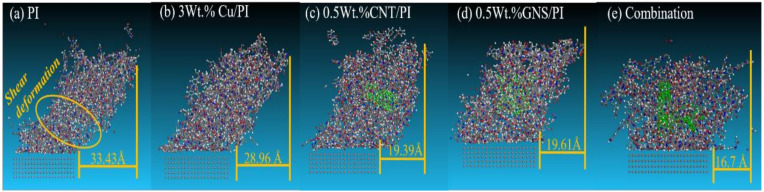
Model of the shear deformations of (**a**) PI, (**b**) 3 wt.% Cu/PI, (**c**) 0.5 wt.% GNS/PI, (**d**) 0.5 wt.% CNT/PI, and (**e**) Combination.

**Figure 11 micromachines-14-00616-f011:**
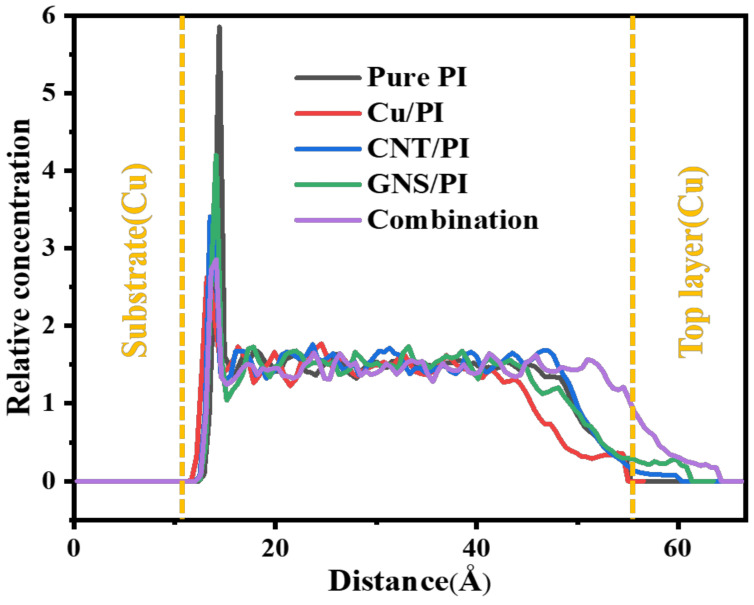
Concentration profiles of friction models.

**Figure 12 micromachines-14-00616-f012:**
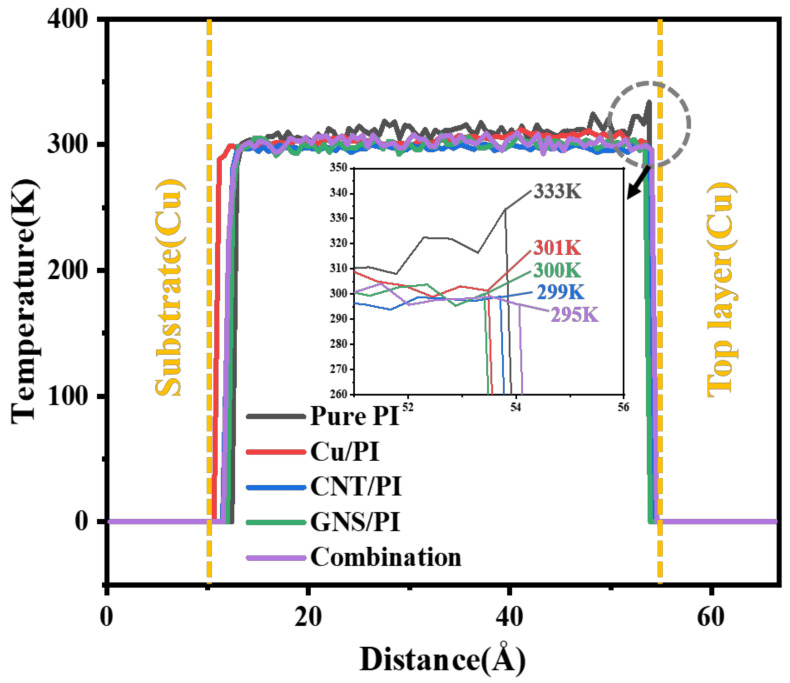
Temperature profiles of friction models.

**Figure 13 micromachines-14-00616-f013:**
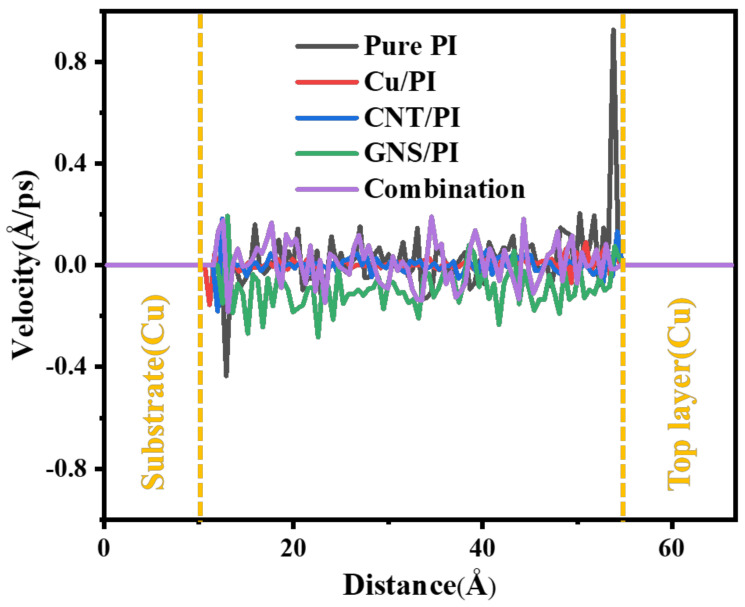
Velocities profiles of friction models.

**Table 1 micromachines-14-00616-t001:** Parameters of optimizations.

Optimization Process	Algorithm	Convergence Criterion	Temperature	Time
Geometry optimization	Smart	2 × 10^−5^ kcal/mol 1 × 10^4^ kcal/mol/Å	/	/
Anneal (NVT)	Nose thermostat	/	300–600 K	2000 ps
NPT	Berendsen barostat	/	298 K	2000 ps

## Data Availability

Where no new data were created.

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
