# Peer review of "Improving Thermal Conductivity and Tribological Performance of Polyimide by Filling Cu, CNT, and Graphene"

_micromachines, 2023, doi:10.3390/mi14030616_

Round 1
Reviewer 1 Report
The authors report in this manuscript that filling Cu, CNT and Graphene can improve the thermal conductivity and tribological performance of polyimide. It’s predictable in theory. Interestingly, this paper uses MD to simulate their performance and explore the improved mechanism from an atomic level, which can provide detailed information about intermolecular interactions and molecular kinetics. This method can also provide guidance on the mechanism exploration for other composites from an atomic level. It can be accept after addressing following problems.
1. In the second paragraph of introduction, please delete the detailed date of fillers except providing references. It’s well known that such fillers have good thermal conductivity.
2. You introduced that Cu, CNT and graphene can improve the thermal conductivity of polymers in experiment. The key point is the proportion, which can form heat conduction channel. However, did you consider the effect of filler content on the thermal conductivity in your study?
3. In section 2.2, please delete the description of the optimization process which was repetitive with Table 1.
4. Please provide the actual model of all materials in Figure 4, especially composite structure. Otherwise, the reader cannot see the distribution of fillers in the PI matrix.
5. In the analysis of Figure 6a, please simplify the reasons from Ref. 25.
6. Why the thermal conductivity of combination is lower than CNT/PI and GNS /PI in Figure 8?
Reviewer 2 Report
This work studied the thermal conductivity, mechanical properties, and tribological properties of polyimide 9 (PI) composites filled by Copper (Cu), carbon nanotube (CNT), and graphene nanosheets (GNS). The effects of copper powder, graphene, and carbon nanotubes on the thermal conductivity of PI were studied by molecular dynamics simulation.
The scope of the work and the reported simulation seem interesting. Having an accurate model would be highly beneficial in reducing the experimental cost. However, increased data points as well as the addition of a hybrid filler option would significantly improve the manuscript's impact. Therefore, I recommend this manuscript for publication after major revision by addressing the following comments:
1- There are minor grammar errors. A full grammar check is highly suggested. Abstract, page 1, line 9: “The thermal conductivity, mechanical properties and tribological properties of polyimide 9 (PI) composites filled by Copper (Cu), carbon nanotube (CNT) and graphene nanosheets (GNS) 10 were investigated by molecular dynamics simulation (MD).” a “,” is required before “and”. E.g., “properties, and tribological” or “(CNT), and graphene”. Also some sentences are too long and the phrase “and” is used multiple time.
2- This work only studied the simulation. However, to evaluate the simulation performance, the models need to be compared with experimental results. The experimental data information is missing.
3- An accurate model would greatly reduce the time and cost of experimental analysis. However, one of the major challenges is to estimate the performance of hybrid filler composites, as optimizing the filler content fraction would significantly increase the data size. Therefore, this work will be greatly improved by studying the performance of hybrid filled (2 or more fillers) PI composites. The synergistic effect of two or more fillers may significantly improve the thermal pathways formation.
4- Another challenge in polymer nanocomposite fabrication would be the dispersion of nanoparticles. This work did not discuss the morphological parameters and the dispersion of the particles compared to the experimental results.
5- Figures 5, 6, and 7, as well as some other figures and tables, do not have the right unit. Please use a complete unit of Wt.% or Vol.% to avoid confusion.

Round 2
Reviewer 2 Report
The revised version of the Manuscript can be accepted after a minor spell check.